# DBGSL: Dynamic Brain Graph Structure Learning

**Alexander Campbell**[1,2]                                   AJRC4@CL.CAM.AC.UK
**Antonio Giuliano Zippo**[3]
**Luca Passamonti**[1]
**Nicola Toschi**[4,5]
**Pietro Liò**[1]

[1] *Department of Computer Science and Technology, University of Cambridge, United Kingdom*

[2] *The Alan Turing Institute, United Kingdom*

[3] *Institute of Molecular Bioimaging and Physiology, National Research Council, Italy*

[4] *University of Rome Tor Vergata, Italy*

[5] *A.A. Martinos Center for Biomedical Imaging, Harvard Medical School, United States*

**Editors:** Accepted for publication at MIDL 2023

## Abstract

Recently, graph neural networks (GNNs) have shown success at learning representations of brain graphs derived from functional magnetic resonance imaging (fMRI) data. The majority of existing GNN methods, however, assume brain graphs are static over time and the graph adjacency matrix is known prior to model training. These assumptions are at odds with neuroscientific evidence that brain graphs are time-varying with a connectivity structure that depends on the choice of functional connectivity measure. Noisy brain graphs that do not truly represent the underling fMRI data can have a detrimental impact on the performance of GNNs. As a solution, we propose Dynamic Brain Graph Structure Learning (DBGSL), a novel method for learning the optimal time-varying dependency structure of fMRI data induced by a downstream prediction task. Experiments demonstrate DBGSL achieves state-of-the-art performance for sex classification using real-world resting-state and task fMRI data. Moreover, analysis of the learnt dynamic graphs highlights prediction-related brain regions which align with existing neuroscience literature. Code available at https://github.com/ajrcampbell/dynamic-brain-graph-structure-learning.

**Keywords:** Dynamic graph, graph neural network, functional magnetic resonance imaging

## 1. Introduction

Functional magnetic resonance imaging (fMRI) is primarily used to measure blood-oxygen level dependent (BOLD) signal in the brain (Huettel et al., 2004). It is one of the most commonly used non-invasive imaging techniques for investigating brain function. Typically, this is accomplished by using a statistical measure of pairwise dependence (e.g. Pearson correlation) to summarize the functional connectivity (FC) between BOLD signals of anatomically separated brain regions (Friston, 1994). The resulting FC matrices (or functional connectomes) have been widely used in graph-based network analysis to understand how the brain works (Sporns, 2022).

Graph neural networks (GNNs) are a type of deep neural network capable of learning representations of graph-structured data (Wu et al., 2020a). By taking FC matrices to represent brain graphs, GNNs have shown recent success at fMRI-related prediction tasks

ranging from phenotypes such as sex (Azevedo et al., 2022) and age (Gadgil et al., 2020), cognitive tasks (Zhang et al., 2021), and brain disorders (Li et al., 2021).

The majority of existing GNN methods applied to fMRI data, however, make two key assumptions: (1) brain graphs are static (i.e. not time-varying), and (2) the true dependency structure between brain regions is known. Although convenient, both assumptions are at odds with a growing body of neuroscientific evidence that FC dynamically changes over time (Calhoun et al., 2014), and that no one statistical measure of dependency exists for truly capturing FC (Mohanty et al., 2020). To ensure that GNNs are able to learn useful representations for use in downstream tasks, it is of high priority to establish the most appropriate way to construct dynamic graphs that best reflect the underling fMRI data.

**Contributions** As a solution, we propose **D**ynamic **B**rain **G**raph **S**tructure **L**earning (DBGSL), the first end-to-end trainable GNN-based model able to learn task-specific dynamic brain graphs from fMRI data in a supervised fashion. Specifically, DBGSL constructs dynamic graph adjacency matrices using spatially attended brain region embeddings learnt from windowed BOLD signals. In addition, DBGSL leverages temporal attention and learnable edge sparsity to further improve classification performance and interpretability. DBGSL achieves state-of-the-art performance for the task of sex classification using real-world resting-state and task fMRI data. Finally, an analysis of the learnt dynamic graphs highlights prediction-related brain regions which align with existing neuroscience literature. Code available on GitHub[1].

## 2. Related work

**Brain graph classification** Recent brain graph classification methods tend to be GNN-based and use static measures of FC for graph construction (Azevedo et al., 2022; Kim and Ye, 2020; Li et al., 2021). To incorporate temporal dynamics, Gadgil et al. (2020) propose a variant of the spatial-temporal GNN (Yan et al., 2018) for fMRI data. However, only graph node features are time-varying whilst the adjacency matrix remains static. More recently, Kim et al. (2021) leverage spatial-temporal attention within a transformer framework (Vaswani et al., 2017) to classify dynamic brain graphs. However, the graph adjacency matrix is assumed to be unweighted, and similar to previous methods, still requires careful selection of a FC measure.

**Graph structure learning** Several graph structure learning (GSL) methods (Zhu et al., 2021; Kalofolias et al., 2017) have been proposed for learning the dependency structure of a dataset, particularly multivaraite timeseries (Cao et al., 2020; Wu et al., 2020b). Unfortunately, a single static graph is usually learnt for all samples making these methods unsuitable for multi-subject fMRI data where differences in subject-level brain graphs are known to be discriminative (Finn et al., 2015). The GSL-based methods proposed in Mahmood et al. (2021), Riaz et al. (2020), and (Kan et al., 2022) are all fMRI specific and learn brain graphs from BOLD signals for use in downstream classification tasks. However, unlike DBGSL, these previous methods assume brain graphs are static.

---

1. https://github.com/ajrcampbell/dynamic-brain-graph-structure-learning

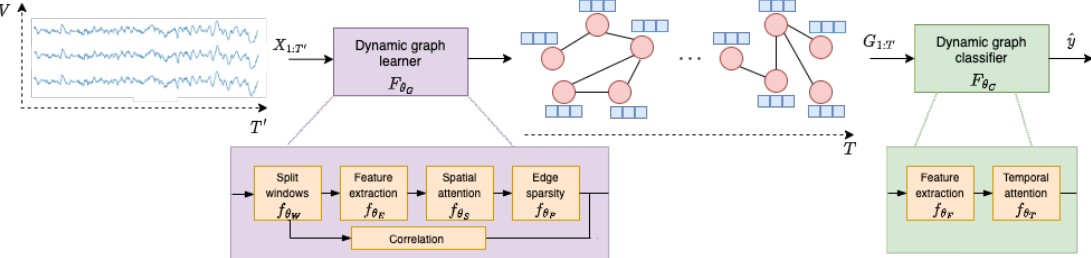

Figure 1: The conceptual framework of DBGSL. Dynamic graph learner $G_{1:T} = (\mathbf{A}_{1:T}, \mathbf{F}_{1:T}) = F_{\theta_G}(\mathbf{X}_{1:T'}) = f_{\theta_W}(f_{\theta_E}(f_{\theta_S}(f_{\theta_P}(\mathbf{X}_{1:T'}))))$. Dynamic graph classifier $\hat{y} = F_{\theta_C}(G_{1:T}) = f_{\theta_T}(f_{\theta_F}(G_{1:T}))$.

## 3. Problem formulation

We formulate dynamic brain GSL in terms of a supervised multivariate timeseries classification problem. Let $\mathbf{X}_{1:T'} = (\mathbf{x}_1, \ldots, \mathbf{x}_{T'}) \in \mathbb{R}^{V \times T'}$ denote BOLD signals from $V$ brain regions measured over $T'$ timepoints and $y \in [0, \ldots C-1]$ a corresponding class label. We assume $\mathbf{X}_{1:T'}$ has a true but unknown nonstationary dependency structure. By letting each brain region correspond to a graph node, we summarize this dependency structure as a dynamic brain graph $G_{1:T} = (\mathbf{A}_{1:T}, \mathbf{F}_{1:T})$ consisting of a dynamic adjacency matrix $\mathbf{A}_{1:T} \in \mathbb{R}_{\geq 0}^{V \times V \times T}$ and dynamic node feature matrix $\mathbf{F}_{1:T} \in \mathbb{R}^{V \times B \times T}$ over $T \leq T'$ snapshots.

Given a dataset $\mathcal{D} \subset \mathcal{X} \times \mathcal{Y}$ consisting of $N$ subjects data $(\mathbf{X}_{1:T'}, y) \in \mathcal{D}$, we aim to train a model $F_\theta(\cdot) = F_{\theta_G} \circ F_{\theta_C}(\cdot)$ with parameters $\theta = \theta_G \cup \theta_C$ that can predict class labels $\hat{y}$ given input $\mathbf{X}_{1:T'}$ using an intermediary learnt dynamic brain graph i.e. $F_\theta(\mathbf{X}_{1:T'}) = F_{\theta_C}(F_{\theta_G}(\mathbf{X}_{1:T'})) = F_{\theta_C}(G_{1:T}) = \hat{y}$. Training consists of minimizing the discrepancy between the actual label $y$ and the predicted label $\hat{y}$, described by a loss function $\mathcal{L}(y, \{\hat{y}, G_{1:T}\})$. The optimization objective is therefore $\theta^* = \arg\min_\theta \mathbb{E}_{(\mathbf{X}_{1:T'}, y) \in \mathcal{D}}[\mathcal{L}(y, F_\theta(\mathbf{X}_{1:T'}))]$.

## 4. Method

As shown in Figure 1, DBGSL consists of two main components: (1) a dynamic graph learner $F_{\theta_G} : \mathcal{X} \to \mathcal{G}$, and (2) a dynamic graph classifier $F_{\theta_C} : \mathcal{G} \to \mathcal{Y}$. We henceforth expand upon the architecture of the dynamic graph learner (Section 4.1), the dynamic graph classifier (Section 4.2), and set-out the training objective (Section 4.3).

### 4.1. Dynamic graph learner

**Split windows**   The dynamic graph learner maps BOLD signals onto a dynamic brain graph such that $F_{\theta_G}(\mathbf{X}_{1:T'}) = G_{1:T}$. To do this, $\mathbf{X}_{1:T'}$ is first split into windows using a temporal-splitting stack transformation $f_{\theta_W}(\cdot)$ following

$$f_{\theta_W}(\mathbf{X}_{1:T'}) = \tilde{\mathbf{X}}_{1:T} = (\tilde{\mathbf{X}}_1, \ldots \tilde{\mathbf{X}}_T), \quad \tilde{\mathbf{X}}_t = \mathbf{X}_{tS:tS+P}, \quad t = 1, \ldots, T \tag{1}$$

where $P$ and $S$ are hyperparameters specifying window length and stride, respectively, $\tilde{\mathbf{X}}_{1:T} \in \mathbb{R}^{P \times V \times T}$ and $T = \lfloor (T' - 2(P-1) - 1)/S + 1 \rfloor$. The hyperparameters are chosen

such that each $\tilde{\mathbf{X}}_t^\top \in \mathbb{R}^{V \times P}$ has a stationary dependency structure within the window of length $P$ timepoints. We leave more data driven methods for selecting $P$, such as statistical tests for stationary (Dickey and Fuller, 1981), as future work.

**Temporal feature extraction** Next, the windowed BOLD signals are input to a 2D convolutional neural network (CNN) $f_{\theta_E}(\cdot)$ in order to extract $K_E$-dimensional feature embeddings for each brain region independently such that $f_{\theta_E}(\tilde{\mathbf{X}}_{1:T}) = \mathbf{H}_{1:T}^G \in \mathbb{R}^{K_E \times V \times T}$. To achieve this, we implement $f_{\theta_E}(\cdot)$ as a inception temporal convolutional network (I-TCN) adapted from Wu et al. (2020b). Specifically, our version of I-TCN takes dilated convolutional kernels and causal padding from original temporal convolutional network (Bai et al., 2018) and combines it with a multi-channel feature extraction in a inception structure (Szegedy et al., 2015). More formally, suppose $f_{\theta_E}(\cdot)$ consists of $L_G$ layers, then for the $l$-th layer with $M$ convolutional filters we have

$$\mathbf{H}_{1:T}^{(l)} = \text{ReLU}\big(\text{BatchNorm}\big(\mathbf{H}_{1:T}^{(l-1)} + ||_{m=1}^M \mathbf{H}_{1:T}^{(l-1)} *_d \mathbf{W}_m^{(l)}\big)\big) \tag{2}$$

where $\mathbf{H}_{1:T}^{(l-1)}, \mathbf{H}_{1:T}^{(l)} \in \mathbb{R}^{K_E \times V \times T}$ are input and output embeddings, respectively, $\mathbf{W}_m^{(l)} \in \mathbb{R}^{\lfloor K_E/M \rfloor \times K_E \times 1 \times S_m}$ is the $m$-th 2D convolutional filter, and $\mathbf{H}_{1:T}^{(0)} = \tilde{\mathbf{X}}_{1:T}$, $\mathbf{H}_{1:T}^{(L_G)} = \mathbf{H}_{1:T}^G$. The symbols $||$ and $*_d$ denote concatenation along the feature dimension and convolution operator with dilation factor $d > 0$, respectively. Following Oord et al. (2016), we set $d = 2^{l-1}$ to exponentially increase the receptive field size of each convolutional kernel with the number of layers and enforce $S_1 < \cdots < S_M$ to allow simultaneous use of small/large kernel lengths to extract short/long-term temporal patterns within a single layer.

**Dynamic adjacency matrix** Since the feature extractor learns embeddings for each brain region independently, to account for spatial relationships we use a self-attention mechanism $f_{\theta_S}(\cdot)$. Specifically, at each snapshot we use the embedding $\mathbf{H}_t^G \in \mathbb{R}^{V \times K_E}$ to learn the dependency structure between brain regions using a simplified version of scaled dot-product self-attention (Vaswani et al., 2017) following

$$\mathbf{A}_t = f_{\theta_S}(\mathbf{H}_t^G) = \text{Sigmoid}\left(\frac{\mathbf{Q}_t \mathbf{K}_t^\top}{\sqrt{K_S}}\right), \quad \mathbf{Q}_t = \mathbf{H}_t^G \mathbf{W}_Q, \quad \mathbf{K}_t = \mathbf{H}_t^G \mathbf{W}_K \tag{3}$$

where $\mathbf{Q}_t, \mathbf{K}_t \in \mathbb{R}^{N \times K_S}$ denote query and key matrices, respectively, which are calculated via $K_S$-dimensional linear projections using trainable matrices $\mathbf{W}_Q, \mathbf{W}_K \in \mathbb{R}^{K_E \times K_S}$. We take each $\mathbf{A}_t$ to be a brain graph adjacency matrix, which by definition is weighted and directed. To make $\mathbf{A}_t$ undirected, which is commonly assumed for brain graph analysis (Friston, 1994), we simply fix $\mathbf{W}_Q = \mathbf{W}_K$. Computing self-attention matrices over the entire sequence of feature embeddings results in a dynamic adjacency matrix $\mathbf{A}_{1:T} \in (0,1)^{V \times V \times T}$ summarizing dynamic FC between brain regions.

**Edge sparsity** By definition, each dynamic adjacency matrix $\mathbf{A}_{1:T}$ is a fully-connected graph at every snapshot. Not only does this make the learnt adjacency matrices difficult to interpret, but it also makes the application of GNNs for learning downstream tasks computationally expensive and susceptible to noise. To tackle this issue, we propose a version of the soft threshold operator (Donoho, 1995) $f_{\theta_P}(\cdot)$ to enforce edge sparsity following

$$f_{\theta_P}(a_{i,j,t}) = \text{ReLU}(a_{i,j,t} - \text{Sigmoid}(\theta_P)), \quad \forall a_{i,j,t} \in \mathbf{A}_{1:T} \tag{4}$$

where $\text{Sigmoid}(\theta_P) \in (0, 1)$ specifies a learnable edge weight threshold. Clearly when $a_{i,j,t} \leq \text{Sigmoid}(\theta_p)$ then $f_{\theta_P}(a_{i,j,t}) = 0$. To ensure the threshold $\text{Sigmoid}(\theta_P)$ starts close to 0 we initialize $\theta_P \approx -10$ so that $\mathbf{A}_{1:T}$ is not mode overly sparse too early on during training.

**Dynamic node feature matrix**  For dynamic node features $\mathbf{F}_{1:T}$ we take the windowed timeseries and compute a (sample) correlation matrix at each snapshot following $\mathbf{F}_t = \tilde{\mathbf{D}}_t^{-1} \boldsymbol{\Sigma}_t \tilde{\mathbf{D}}_t^{-1}$ where $\tilde{\mathbf{D}}_t = \sqrt{\text{diag}(\boldsymbol{\Sigma}_t)}$ and $\boldsymbol{\Sigma}_t = \frac{1}{P-1} \tilde{\mathbf{X}}_t^\top (\mathbf{I}_P - \frac{1}{P} \mathbf{1}_P^\top \mathbf{1}_P) \tilde{\mathbf{X}}_t$ with $\mathbf{I}_P$ and $\mathbf{1}_P$ being a $P \times P$ identity matrix and $1 \times P$ matrix of all ones, respectively. This choice is motivated by previous work on static brain graphs where a node's FC profile achieves superior performance over other features (Li et al., 2021; Kan et al., 2022; Cui et al., 2022)

## 4.2. Dynamic graph classifier

**Spatial-temporal feature extraction**  We next use a $L_C$-layered recurrent GNN $f_{\theta_F}(\cdot)$ similar to Seo et al. (2018) to learn a spatial-temporal representation of $G_{1:T}$. For simplicity, we implement the recurrent mechanism using a gated recurrent unit (GRU) (Cho et al., 2014) and each gate as a graph convolutional network (GCN) (Kipf and Welling, 2016). Specifically, the GCN for each gate is defined $\text{GCN}(\mathbf{F}_t, \mathbf{A}_t) = \hat{\mathbf{D}}_t^{-1/2} \hat{\mathbf{A}}_t \hat{\mathbf{D}}_t^{-1/2} \mathbf{F}_t \mathbf{W}_1$ where $\mathbf{W}_1 \in \mathbb{R}^{K_C \times K_C}$ is a trainable weight matrix and $\hat{\mathbf{D}}_t = \text{diag}(\hat{\mathbf{A}}_t \mathbf{1}_{1 \times V}^\top)$ is the degree matrix with $\hat{\mathbf{A}}_t = \mathbf{A}_t + \mathbf{I}_V$. The $l$-th layer of the GRU at each snapshot is then described following

$$\mathbf{R}_t^{(l)} = \text{Sigmoid}(\text{GCN}(\tilde{\mathbf{H}}_t^{(l-1)} || \tilde{\mathbf{H}}_{t-1}^{(l)}, \mathbf{A}_t)), \quad \mathbf{U}_t^{(l)} = \text{Sigmoid}(\text{GCN}(\tilde{\mathbf{H}}_t^{(l-1)} || \tilde{\mathbf{H}}_{t-1}^{(l)}, \mathbf{A}_t)) \quad (5)$$

$$\mathbf{C}_t^{(l)} = \text{Tanh}(\text{GCN}(\tilde{\mathbf{H}}_t^{(l-1)} || \mathbf{R}_t^{(l)} \odot \tilde{\mathbf{H}}_{t-1}^{(l)}, \mathbf{A}_t)), \quad \tilde{\mathbf{H}}_t^{(l)} = \mathbf{U}_t^{(l)} \odot \tilde{\mathbf{H}}_{t-1}^{(l)} + (1 - \mathbf{U}_t^{(l)}) \odot \mathbf{C}_t^{(l)} \quad (6)$$

where $\mathbf{R}_t^{(l)}, \mathbf{U}_t^{(l)} \in \mathbb{R}^{V \times K_C}$ are the reset and update gates, respectively, and $\tilde{\mathbf{H}}_t^{(l)} \in \mathbb{R}^{V \times K_C}$ is the hidden state such that $\tilde{\mathbf{H}}_t^{(0)} = \mathbf{F}_t$ with $\tilde{\mathbf{H}}_0^{(l)} \in \mathbf{0}_{V \times K_C}$ being initialized as a matrix of zeros. The symbols $\odot$ and $||$ denote the Hadamard product and the feature-wise concatenation operator, respectively. Iterating through (5)-(6) for each graph snapshot, we end up with per-layer output embeddings $\mathbf{H}_{1:T}^{(l)} \in \mathbb{R}^{V \times K_C \times T}$ which we concatenated along the feature dimension, to combine information from neighbors that are up to $L_C$-hops away from each node, and then averaged over the node dimension to create a sequence of brain graph embeddings denoted $\mathbf{H}_{1:T}^C = \boldsymbol{\phi}(||_{l=1}^{L_C} \mathbf{H}_{1:T}^{(l)}) \in \mathbb{R}^{K_C L_C \times T}$ where $\boldsymbol{\phi} = \frac{1}{V} \mathbf{1}_{1 \times V}$ is an average pooling matrix.

**Temporal attention readout**  To emphasize snapshots with the most important brain graph embeddings, we next employ a novel temporal attention readout layer $f_{\theta_T}(\cdot)$ adapted from squeeze-and-excite attention networks (Hu et al., 2018). More formally, we define a temporal attention score matrix $\boldsymbol{\alpha} \in (0, 1)^{1 \times T}$ following

$$\boldsymbol{\alpha} = \text{Sigmoid}(\text{ReLU}(\boldsymbol{\psi} \mathbf{H}_{1:T}^C \mathbf{W}_2) \mathbf{W}_3) \quad (7)$$

where $\mathbf{W}_2 \in \mathbb{R}^{T \times \tau T}$, $\mathbf{W}_3 \in \mathbb{R}^{\tau T \times T}$ are trainable weight matrices that encode temporal dependencies via a bottleneck controlled by the hyperparameter $\tau \in (0, 1]$ and $\boldsymbol{\psi} = \frac{1}{K_C L_C} \mathbf{1}_{1 \times K_C L_C}$. The final graph-level representation $\mathbf{h}_\mathcal{G} \in \mathbb{R}^{L_C K_C}$ is obtained using the temporal attention score matrix to take the weighted sum over snapshots following $\mathbf{h}_\mathcal{G} = (\boldsymbol{\alpha} \odot \mathbf{H}_{1:T}^C) \boldsymbol{\xi}^\top$ where $\xi = \mathbf{1}_{1 \times T}$ is a sum pooling matrix. The representation is then passed through a linear layer mapping it onto unscaled log probabilities $\log p(y|\mathbf{X}_{1:T'}) \in \mathbb{R}^C$.

## 4.3. Loss function

We train DBGSL by minimizing cross-entropy loss $\mathcal{L}_{\text{CE}}(y, \hat{y})$ as well as a collection of prior constraints on the learnt graphs denoted $\mathcal{R}(G_{1:T})$ such that $\mathcal{L}(y, \{\hat{y}, G_{1:T}\}) = \mathcal{L}_{\text{CE}}(y, \hat{y}) + \mathcal{R}(G_{1:T})$ where $\mathcal{L}_{\text{CE}}(y, \hat{y}) = -\sum_{c=1}^{C} \mathbb{1}(y = c) \log p(y|\mathbf{X}_{1:T'})_c$. This encourages DBGSL to learn task-aware dynamic graphs that encode interpretable class differences into $\mathbf{A}_{1:T}$.

**Regularization constraints** Since connected nodes in a graph are more likely to share similar features (McPherson et al., 2001) we add a regularization term encouraging feature smoothness of the learnt graphs defined as $\mathcal{L}_{\text{FS}}(\mathbf{A}_{1:T}, \mathbf{F}_{1:T}) = \frac{1}{V^2} \sum_{t=1}^{T} \text{Tr}(\mathbf{F}_t^\top \hat{\mathbf{L}}_t \mathbf{F}_t)$ where $\text{Tr}(\cdot)$ denotes the matrix trace operator and $\hat{\mathbf{L}}_t = \mathbf{D}_t^{-1/2} \mathbf{L}_t \mathbf{D}_t^{-1/2}$ is the (symmetric) normalized Laplacian matrix defined as $\mathbf{L}_t = \mathbf{D}_t - \mathbf{A}_t$ where $\mathbf{D}_t = \text{diag}(\mathbf{A}_t \mathbf{1}_{V \times 1})$ which makes feature smoothness node degree independent (Ando and Zhang, 2006). Furthermore, to discourage volatile changes in graphs between timepoints we also add a prior constraint encouraging temporal smoothness defined as $\mathcal{L}_{\text{TS}}(\mathbf{A}_{1:T}) = \sum_{t=1}^{T-1} ||\mathbf{A}_t - \mathbf{A}_{t+1}||_1$ where $|| \cdot ||_1$ denotes the matrix L1-norm. Moreover, to encourage the learning of a large sparsity parameter $\text{Softmax}(\theta_P)$ in (4), we further add a sparsity regularization term defined $\mathcal{L}_{\text{SP}}(\mathbf{A}_{1:T}) = \sum_{t=1}^{T} ||\mathbf{A}_t||_1$. In combination with $\mathcal{L}_{\text{CE}}(\cdot, \cdot)$, this ensures only the most import task-specific edges are kept in $\mathbf{A}_{1:T}$. The final loss function we seek to minimize is

$$\mathcal{L}(y, \{\hat{y}, G_{1:T}\}) = \mathcal{L}_{\text{CE}}(y, \hat{y}) + \lambda_{\text{FS}} \mathcal{L}_{\text{FS}}(\mathbf{F}_{1:T}, \mathbf{A}_{1:T}) + \lambda_{\text{TS}} \mathcal{L}_{\text{TS}}(\mathbf{A}_{1:T}) + \lambda_{\text{SP}} \mathcal{L}_{\text{SP}}(\mathbf{A}_{1:T}) \quad (8)$$

where $\lambda_{\text{FS}}, \lambda_{\text{TS}}, \lambda_{\text{SP}} \geq 0$ are hyperparameters weighting regularization contributions.

## 5. Experiments

We evaluate the performance of DBGSL on the task of biological sex classification, a widely used benchmark for supervised deep learning-based models designed for fMRI data (Kim et al., 2021; Gadgil et al., 2020; Azevedo et al., 2022). Biological sex differences in the brain are supported by a large body of neuroscience literature (Bell et al., 2006; Mao et al., 2017).

**Datasets** We construct two datasets using publicly available fMRI data from the Human Connectome Project (HCP) (Van Essen et al., 2013). The first dataset consists of resting-state fMRI data from $N = 1,095$ subjects with $T' = 1,200$ (HCP-Rest). The second dataset consists of task fMRI data from $N = 926$ subjects performing the emotional task with $T' = 176$ (HCP-Task). We parcellate the fMRI data into $V = 243$ region-wise BOLD signals using the Brainnetome atlas (Fan et al., 2016). The biological sex of each subject is taken as a class label $C = 2$. We refer to Appendix A for further details on each dataset.

**Baselines** We compare DBGSL against a range of baselines broadly grouped by whether they take as input static/dynamic brain graphs or region-wise BOLD signals. For static (linear) baselines, we include kernel ridge regression (KRR) (He et al., 2020) and support vector machine (SVM) (Abraham et al., 2017). For static deep learning baselines we include a multilayer perception (MLP) and BrainnetCNN (BNCNN) (Kawahara et al., 2017) where for dynamic baselines we include ST-GCN (STGCN) (Gadgil et al., 2020) and STA-GIN (Kim et al., 2021). For GSL baselines, we include FBNetGen (FBNG) (Kan et al., 2022) and Deep fMRI (DFMRI) (Riaz et al., 2020). Finally, we include a bidirectional

LSTM (BLSTM) (Hebling Vieira et al., 2021) which learns directly from BOLD signals. Further details about each baseline model can be found in Appendix B.

**Implementation** We split both datasets into 80/10/10% training/validation/test data maintaining class proportions. For fairness of comparison, all models are trained using the Adam optimizer (Kingma and Ba, 2014) with decoupled weight decay (Loshchilov and Hutter, 2017). We use the hyperparameter settings described in the original implementation of each baseline model as the starting point for tuning. We train all model for $5,000$ epochs using early-stopping with a patience of 15 based on the lowest accuracy on the validation dataset. Finally, we train all models 5 times using different random seeds as well as dataset splits. Further implementation details can be found in Appendix C.

**Evaluation metrics** Overall performance is evaluated using mean test accuracy (ACC) and area under the receiver operator curve (AUROC). For model comparisons, we use the almost stochastic order (ASO) test (Del Barrio et al., 2018; Dror et al., 2019) with significance level $\alpha = 0.05$ and correct for multiple comparisons (Bonferroni, 1936).

**Results** Sex classification results are summarized in Table 1. Clearly DBGSL is the best performing model across both datasets as measured by accuracy and AUROC. On HCP-Rest, DBGSL outperforms the second-best baseline KRR in terms of accuracy by 8.82 percentage points (pp) whereas on HCP-Task the second-best baseline SVM is outperformed by 8.17 pp, with both gains being statistically significant. Similar to the findings of He et al. (2020), when considering only static brain graphs the linear baselines KRR and SVM either outperform each deep learning baseline or the difference in result is not statistically significant. We attribute the superior performance of DBGSL to the brain graph being learnt rather than fixed prior to training as well as incorporating temporal dynamics. For further analysis (including ASO test scores), we refer to Appendix D.

Table 1: Sex classification results for HCP-Rest and HCP-Task (mean plus/minus standard deviation across five runs). First and second-best results are **bold** and underlined, respectively. Statistically significant difference from DBGSL marked *.

| Model | HCP-Rest | | HCP-Task | |
|---|---|---|---|---|
| | ACC (%, ↑) | AUROC (↑) | ACC (%, ↑) | AUROC (↑) |
| KRR | 83.50 ± 1.94 * | 0.9187 ± 0.0025 * | 81.37 ± 2.17 * | 0.9031 ± 0.0185 * |
| SVM | 82.70 ± 2.68 * | 0.9170 ± 0.0089 * | 83.16 ± 1.91 * | 0.9097 ± 0.0184 * |
| MLP | 81.47 ± 3.29 * | 0.9091 ± 0.0281 * | 81.10 ± 3.44 * | 0.8837 ± 0.0250 * |
| BLSTM | 81.50 ± 1.26 * | 0.9058 ± 0.0081 * | 77.24 ± 4.05 * | 0.8526 ± 0.0188 * |
| BNCNN | 76.83 ± 7.46 * | 0.6156 ± 0.0837 * | 70.66 ± 8.23 * | 0.5945 ± 0.0499 * |
| STGCN | 62.63 ± 4.50 * | 0.6991 ± 0.0264 * | 54.87 ± 3.37 * | 0.5629 ± 0.0355 * |
| DFMRI | 82.65 ± 3.40 * | 0.8941 ± 0.0342 * | 81.34 ± 2.19 * | 0.8024 ± 0.0317 * |
| FBNG | 81.57 ± 2.90 * | 0.8967 ± 0.0170 * | 77.16 ± 3.90 * | 0.8548 ± 0.0320 * |
| STAGIN | 83.13 ± 2.11 * | 0.8597 ± 0.0467 * | 81.88 ± 2.73 * | 0.8088 ± 0.0404 * |
| DBGSL | **92.32 ± 2.22** | **0.9623 ± 0.0433** | **89.54 ± 3.48** | **0.9496 ± 0.0423** |

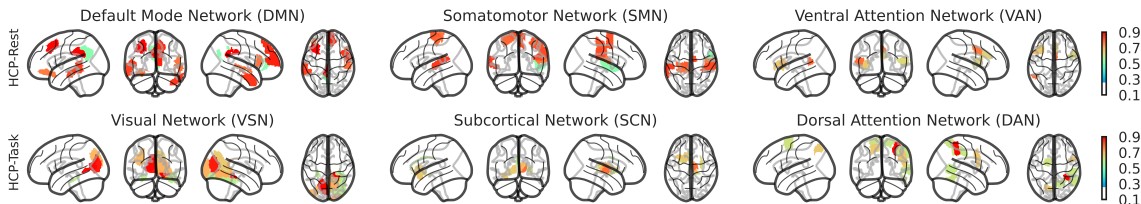

Figure 2: Sex-discriminative brain region scores **z** (normalized to $[0, 1]$) for HCP-Rest (top) and HCP-Task (bottom).

## 6. Interpretability analysis

A major strength of DBGSL is it's ability to learn task-ware dynamic brain graph structure from BOLD signals which have uses beyond classification. To highlight the brain region(s) that are most sex-discriminative, we create a brain region score vector using temporally weighted node degree $\mathbf{z} = \frac{1}{T} \sum_{t=1}^{T} (\sum_{j=1}^{V} \mathbf{A}_{j,t}) \alpha_t \in \mathbb{R}^V$. We take all regions falling within the top 20% across all subjects in the test dataset and plot them with respect to the intrinsic connectivity networks of Yeo et al. (2011) in Figure 2.

**Results**   For HCP-Rest, 25.5% of the highest scores are within the default mode network (DMN), a key network that is consistently observed in resting-state fMRI studies (Mak et al., 2017; Satterthwaite et al., 2015). Within the DMN the brain regions with the highest sex-prediction ability are localized in the dorsal anterior cingulate cortex, middle frontal gyrus, and posterior superior temporal cortex. These fronto-temporal brain regions are key components of the theory of mind network, which underlies a meta-cognitive function in which females excel (Adenzato et al., 2017). Another key region in theory of mind tasks, the posterior superior temporal cortex is found to reliably predict sex within the ventral attention network (VAN). For HCP-Task, 30.6% of the highest scores are in the parahippocampal gyrus, medial occipital cortex, and superior parietal lobule which form a posterior visual network (VSN). The fact that such regions best discriminated males from females reflects differences in the ability to process emotional content and/or sex-related variability in directing attention to certain features of emotional stimuli (Mackiewicz et al., 2006), like the facial expressions from the HCP task paradigm (Markett et al., 2020). For further analysis we refer to Appendix E.

## 7. Conclusion

We propose **D**ynamic **B**rain **G**raph **S**tructure **L**earning (DBGSL), an end-to-end trainable model capable of learning optimal time-varying dependency structure from fMRI data in the form of a dynamic brain graph. To the best of our knowledge, we are the first to propose and address a dynamic GSL problem via GNN-based deep learning on BOLD signals derived from fMRI data. Central to our approach is the use of spatial-temporal attention to exploit the inherent inter and intra relationships of brain region BOLD signals. Extensive experiments on two real-world fMRI datasets demonstrates that DBGSL achieves state-of-the-art results for sex classification. Future research directions include evaluating DBGSL

on more brain classifications tasks such as age or fluid intelligence, learning the optional window size and stride from the data, and incorporating higher-order brain interactions into the GSL process.

## Acknowledgments

This work is supported by The Alan Turing Institute under the EPSRC grant EP/N510129/1. Data were provided [in part] by the Human Connectome Project, WU-Minn Consortium (Principal Investigators: David Van Essen and Kamil Ugurbil; 1U54MH091657) funded by the 16 NIH Institutes and Centers that support the NIH Blueprint for Neuroscience Research; and by the McDonnell Center for Systems Neuroscience at Washington University.

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

## Appendix A. Datasets

We construct two multivariate timeseries classification datasets using publicly available fMRI scans from the Human Connectome Project (HCP) S1200 release[2] (Van Essen et al.,

---

2. https://db.humanconnectome.org

2013). All HCP data is collected from voluntary healthy participants with informed consent and is fully anonymized. The two datasets differ depending on whether the subjects are resting (HCP-Rest) or performing a specific task (HCP-Task) during the acquisition of the images.

**HCP-Rest**  We consider resting-state fMRI scans minimally pre-processed following the pipeline described in Glasser et al. (2013). A total of $N = 1,095$ subjects are selected from the first scanning-session (of four) using left-right phase encoding. The subjects are instructed to rest for 15 minutes during image acquisition. The repetition time (TR), i.e. the time between successive image acquisitions, is 0.72 seconds resulting in $T' = 1,200$ images per subject. We took the biological sex of each subject as a label making the total number of classes $C = 2$. Female subjects accounted for 54.4% of the total dataset.

**HCP-Task**  We consider task fMRI scans from the emotional task minimally pre-processed following the pipeline described in Glasser et al. (2013). A total of $N = 926$ are selected from the first scanning-session (of two) with left-right phase encoding. For the emotional task, subjects are asked to indicate which of two faces or which of two shapes presented at the bottom of a screen match the face (or shape) at the top of the screen. With TR = 0.72 seconds and a scanning session lasting 2.11 minutes, the number of images per subject is $T' = 176$. Female subjects accounted for 51.2% of the total dataset.

**Further preprocessing**  Since the fMRI scans from both datasets are a timeseries of 3D brain volumes, we parcellate them into $V = 243$ mean brain region (210 cortical, 36 subcortical) BOLD signals of length $T'$ time points using the Brainnetome atlas [3] (Fan et al., 2016). Each timeseries is then transformed into a $z$-score, by standardizing region-wise, in order to remove amplitude effects. Finally, to balance class proportions across both datasets, the minority class is randomly oversampled.

## Appendix B. Baselines

We compare DBGSL against a range of different baseline models, all having been previously used to classify fMRI data with publicly available code. The baselines are broadly grouped by whether they take as input static FC, dynamic FC, or BOLD signals. To make the comparison fair, we include traditional machine learning models as recent studies have shown they achieve comparable results to deep learning-based models on fMRI data (He et al., 2020).

**Kernel ridge regression[4] (KRR)**  (He et al., 2020)  Kernel ridge regression (KRR) combines ridge regression, i.e. linear least squares with L2-norm regularization, with the kernel trick in order to learn a linear function in the space induced by the kernel and the input data (Murphy, 2012a). Following He et al. (2020), we use a linear kernel and keep the weight on the regularization loss as a tunable hyperparameter. As input, KRR takes the vectorized lower-triangle (excluding the principal diagonal) of static FC matrices computed using Pearson correlation.

---

3. https://atlas.brainnetome.org

4. https://github.com/ThomasYeoLab/CBIG/blob/master/stable_projects/predict_phenotypes/He2019_KRDNN/

**Support vector machine (SVM)** (Abraham et al., 2017) Support vector machine learns vectors defining a hyperplane, i.e. a decision boundary, that maximizes the margin between input data from different classes after projection into a higher dimensional space using a kernel function (Murphy, 2012b). Following Abraham et al. (2017), we use a linear kernel and keep the weight on the regularization loss as a tunable hyperparameter. Similar to KRR, SVM takes as input a vectorized static FC matrix computed using Pearson correlation.

**Multilayered perceptron[4] (MLP)** (Hebling Vieira et al., 2021) A multilayered perceptron (MLP) taking as input vectorized static FC matrices computed using Pearson correlation. Used as a baseline in Kawahara et al. (2017) and Gadgil et al. (2020), we follow Hebling Vieira et al. (2021) and implement MLP using three linear layers with dropout, batch normalization, and rectified linear unit (ReLU) activation functions after the first two layers. The hidden dimension in each layer is treated as a tunable hyperparameter.

**Bi-directional long short-term memory (BLSTM)** (Hebling Vieira et al., 2021) A bi-directional long short-term memory (BI-LSTM) recurrent neural network which is able to learn patterns directly from the BOLD signals rather than from precomputed FC matrices. Following Hebling Vieira et al. (2021), we use two bi-directional LSTM layers (Graves and Schmidhuber, 2005) each processing BOLD signals forward in time and backward in time with the hidden representations being combined via addition. The hidden dimension in each layer is kept as a tunable hyperparameter.

**BrainNetCNN[4] (BNCNN)** (Kawahara et al., 2017) A convolutional neural network with specially designed cross convolutional filters, i.e. edge-to-edge and edge-to-node, used for learning topological features directly from static FC matrices taken as input. Originally proposed in Kawahara et al. (2017), we use the implementation from He et al. (2020) with four layers and keep the number of hidden channels in the last layer as a tunable hyperparameter.

**Spatio-temporal graph convolutional network[5] (STGCN)** (Gadgil et al., 2020) A GNN consisting of three spatio-temporal blocks implemented as a GCN layer for extracting spatial features and a 1D convolutional layer for extracting temporal. Node features are taken as windows of BOLD signals and the adjacency matrix is taken to be the average FC matrix, computed using Pearson correlation, over all subjects in the training dataset. We treat the number of hidden features as a hyperparameter to be tuned.

**Deep fMRI (DFMRI)** (Riaz et al., 2020) A deep learning-based GSL method which learns static brain graphs directly from BOLD signals using a 1D CNN feature extractor, a MLP graph constructor, and a MLP graph classifier (Riaz et al., 2020). Inspired by Siamese-networks (Bromley et al., 1993), the graph constructor learns a similarity score between pairs of extracted features from two different brain regions. We treat the hidden dimension in the graph classifier as a tunable hyperparameter.

**Functional brain network generator[6] (FBNG)** (Kan et al., 2022) A GSL method similar to DEEP-FMRI which learns static brain graphs directly from BOLD signals but instead using a LSTM feature extractor and a GNN as a graph classifier (Kan et al., 2022).

---

5. https://github.com/sgadgil6/cnslab_fmri

6. https://github.com/Wayfear/FBNETGEN

Unlike DEEP-FMRI which uses a MLP to learn a graph adjacency matrix, FBNG simply takes the inner product between extracted features. FBNETGEN also introduces a group inter loss that aims to maximize the difference in learnt graphs across different classes, while keeping those within the same class similar. The hidden dimension in graph classifier is treated as a tunable hyperparameter.

**Spatio-temporal attention graph isomorphism network**[7] **(STAGIN)** (Kan et al., 2022) A joint GNN and transformer that takes as input attributed unweighted dynamic graphs derived from sliding window FC. Following Kim et al. (2021), we use Pearson correlation as the measure of FC and binarize the matrices by thresholding the top 30-percentile values as connected. We fix the number of layers in the GNN to four and treat the node embedding dimension as a hyperparameter to be tuned.

## Appendix C. Implementation

**Software and hardware** All models are developed in Python 3.7 (Python Core Team, 2019) using scikit-learn 1.1.1 (Pedregosa et al., 2011), PyTorch (Paszke et al., 2019), and numpy 1.1.1 (Harris et al., 2020). All metrics are implemented using TorchMetrics 1.1.1 (Nicki Skafte Detlefsen et al., 2022) and statistical significance tests are carried out using deepsignificance 1.1.1 (Ulmer et al., 2022). Experiments are performed on a Linux server (Debian 5.10.113-1) with a NVIDIA RTX A6000 GPU with 48 GB memory and 16 CPUs.

**Training and testing** All baselines are implemented as per the original paper and/or code repository given in Appendix B. To ensure differences in classification performance could be attributed as much as possible to differences in model architecture, paper specific training and testing strategies are removed. In particular, during inference for STGCN and BLSTM only a single model is used to make predictions instead of an ensemble of models. Furthermore, we removed mixup/label smoothing from FBNG, and did not use the one-cycle learning rate scheduler for STAGIN. These different training and testing strategies can all be considered types of regularization, the addition of which, would benefit the performance of any model.

**Hyperparameter optimization** We use model and training hyperparameter values described in the original implementation of each baseline as a starting point for hyperparameter tuning on the validation dataset. Since searching for the optional values of hyperparameters for every baseline is outside the scope of the paper, we focus mainly on tuning regularization loss weights (KRR, SVM) and the dimensions of hidden layers (MLP, BLSTM, BNCNN, STGCN, DFMRI, FBNG, STAGIN). For DBGSL, we fix the number of filters in the temporal feature extractor $f_{\theta_E}(\cdot)$ to $M = 3$ and the bottleneck in the temporal attention layers $f_{\theta_T}(\cdot)$ to $\tau = 0.5$. For all other hyperparameters see Table 2.

---

7. https://github.com/egyptdj/stagin

Table 2: Optimal hyperparameter values for DBGSL on HCP-Rest and HCP-Task (based on 5 runs, lowest validation accuracy).

| Hyperparameter | Range | HCP-Rest | HCP-Task |
|---|---|---|---|
| Training | | | |
| - Batch size | {5, 10, 20, 50} | 20 | 20 |
| - Learning rate | {1e-2, 1e-3, 1e-4} | 1e-3 | 1e-3 |
| - Weight decay | {1e-5, 1e-4, 1e-3} | 1e-4 | 1e-4 |
| Model | | | |
| - Dynamic graph learner | | | |
| – Window length, $P$ | {5, 10, 30, 50, 70, 100} | 50 | 30 |
| – Window stride, $S$ | {1, 3, 5, 10, 25, 50} | 3 | 1 |
| – Number of layers, $L_G$ | {1, 2, 3, 4, 5, 6} | 4 | 4 |
| – Number of features, $K_E$ | {8, 16, 32, 128, 256} | 64 | 64 |
| – Filter sizes, $S_m$ | {{3, 5, 7}, {4, 8, 16}} | {4, 8, 16} | {4, 8, 16} |
| – Embedding size $K_S$ | {4, 8, 16, 32, 64, 128} | 16 | 16 |
| - Dynamic graph classifier | | | |
| – Number of layers, $L_C$ | {1, 2, 3, 4, 5, 6} | 3 | 3 |
| – Number of features, $K_C$ | {8, 16, 32, 64, 128, 256} | 64 | 64 |
| - Feature smoothness, $\lambda_{\mathrm{FS}}$ | {1e-4, 1e-3, 1e-2} | 1e-4 | 1e-4 |
| - Temporal smoothness, $\lambda_{\mathrm{TS}}$ | {1e-4, 1e-3, 1e-2} | 1e-3 | 1e-4 |
| - Sparsity, $\lambda_{\mathrm{SP}}$ | {1e-4, 1e-3, 1e-2} | 1e-3 | 1e-3 |

## Appendix D. Experiments

### D.1. Sex classification

Figure 4 shows individual almost stochastic order (ASO) test (Dror et al., 2019)[8] statistics for the sex classification task. The ASO test has been recently proposed to test the statistical significance of deep learning models. In general, the ASO test determines whether a stochastic order (Reimers and Gurevych, 2018) exists between two models based on their respective sets of test dataset scores obtained from multiple runs, i.e., different random seeds. Given test dataset scores of two models $A$ and $B$ over multiple runs, the ASO test computes a test-statistic $\epsilon_{\min}$ that indicates how far model $A$ is from being significantly better than model $B$. When the distance $\epsilon_{\min} = 0.0$, one can claim that model $A$ is stochastically dominant over model $B$, denoted $A \succ B$, with a predefined significance level $\alpha \in (0, 1)$. When $\epsilon_{\min} < 0.5$ one can say model $A$ almost stochastically dominates model $B$, denoted $A \succeq B$. On the other hand, when $\epsilon_{\min} = 1.0$, this means $B \preceq A$. For $\epsilon_{\min} = 0.5$, no order can be determined.

---

8. https://github.com/Kaleidophon/deep-significance

### D.2. Ablation study

We conduct an ablation study to investigate the impact on performance of DBGSL without (w/o) key model components. Specifically, within the dynamic graph learner $F_{\theta_G}(\cdot)$ we replace the I-TCN $f_{\theta_E}(\cdot)$ with a 1D convolution with filter length 4 (w/o inception), replace self-attention $f_{\theta_S}(\cdot)$ with a normalized Person correlation matrix (w/o spatial att.), remove edge sparsity $f_{\theta_P}(\cdot)$ with $\lambda_{SP} = 0$ (w/o sparsity), and remove temporal attention $f_{\theta_T}(\cdot)$ (w/o temporal att.). In addition, we also remove feature smoothness $\lambda_{FS} = 0$ (w/o feature reg.) and temporal smoothness $\lambda_{TS} = 0$ (w/o temporal reg.) graph regularization terms from the loss function.

Table 3: Ablation study results on HCP-Rest and HCP-Task (mean plus/minus standard deviation across five runs). Best results are **bold**. $\Delta$ = percentage point (pp) change, w/o = without model component. Statistically significant difference from DBGSL marked *.

| Model | HCP-Rest | | HCP-Task | |
|---|---|---|---|---|
| | ACC (%, ↑) | $\Delta$ (pp) | ACC (%, ↑) | $\Delta$ (pp) |
| DBGSL | **92.32 ± 2.22** | | **89.54 ± 3.48** | |
| - w/o inception $f_{\theta_E}(\cdot)$ | 91.22 ± 2.69 | ↓ 1.10 * | 88.79 ± 2.34 | ↓ 0.75 |
| - w/o self att. $f_{\theta_S}(\cdot)$ | 89.97 ± 3.04 | ↓ 2.34 * | 86.80 ± 3.76 | ↓ 2.74 * |
| - w/o sparsity $f_{\theta_P}(\cdot)$, $\lambda_{SP} = 0$ | 92.32 ± 2.23 | ↓ 1.04 * | 87.00 ± 2.33 | ↓ 2.54 * |
| - w/o temporal att. $f_{\theta_T}(\cdot)$ | 92.26 ± 2.43 | ↓ 0.06 | 87.56 ± 2.84 | ↓ 1.98 * |
| - w/o feature reg. $\lambda_{FS} = 0$ | 92.29 ± 2.39 | ↓ 0.03 | 88.50 ± 2.87 | ↓ 1.04 |
| - w/o temporal reg. $\lambda_{TS} = 0$ | 92.12 ± 2.21 | ↓ 0.20 | 88.43 ± 3.23 | ↓ 1.11 |

**Results** Table 3 summarizes the ablation study results. Clearly the use of self-attention significantly improves accuracy across both datasets (HCP-Rest ↑ 2.34 pp vs HCP-Task ↑ 2.74 pp) since it allows for task-aware spatial relationships between brain regions to be built into the dynamic graph adjacency matrix for use by the graph classifier. Similarly, sparsity also significantly improves accuracy (HCP-Rest ↑ 1.04 pp vs ↑ 2.54 pp HCP-Task) since it removes noisy edges from the dynamic adjacency matrix thereby reducing errors from being propagated to node representations in the graph classifier. Finally, the effect of the I-TCN is significant for HCP-Rest (↑ 1.10 pp) but only marginal for HCP-Task (↑ 0.75 pp) which might be explained by the fact that the BOLD timeseries from the former dataset are collected over a longer time period then the latter thereby benefiting more from larger kernel sizes being able to extract longer temporal patterns.

### D.3. Hyperparameter sensitivity

We conduct a sensitivity analysis on the main hyperparameters which influence the complexity of DBGSL. In particular, for the dynamic graph learner $F_{\theta_G}(\cdot)$ we vary window length $P$, window stride $S$, embedding size $K_E$, and number of layers $L_G$. On the other hand, for the dynamic graph classifier $F_{\theta_C}(\cdot)$ we vary number of layers $L_C$ and number of

features $K_C$. For each experiment, we change the parameter under investigation and fix other parameters to their optimally tuned values.

Figure 3: Sensitivity analysis results on HCP-Rest and HCP-Task. Results are mean plus/minus standard deviation across five runs.

**Results**   Figure 3 shows the results of the hyperparameter sensitivity analysis. On both HCP-Rest and HCP-Task we see that increasing window length $P$ decreases accuracy which we attribute to the fact that more data within a window makes it harder for the dynamic graph learner to identify fast changes between brain regions that are task discriminative. A similar relationship holds for increasing window stride $S$ due to the loss of information among contiguous regions of BOLD signals when building dynamic graphs. Furthermore, increasing the depth of information propagation beyond 3 hops in the dynamic graph classifier decreases performance as shown by the number of layers $L_C$. Finally, increasing the number of layers $L_G$ and embedding size $K_E$ in the graph learner as well as the number of features $K_C$ in the graph classifier each show diminishing returns for performance gains.

## Appendix E. Interpretability analysis

Figure 5 shows an example dynamic adjacency matrix from a dynamic brain graph learnt by DBGSL $\mathbf{A}_{1:T}$ compared to a dynamic FC matrix calculated using Pearson correlation $\mathbf{A}_{1:T}^C$ following Calhoun et al. (2014).

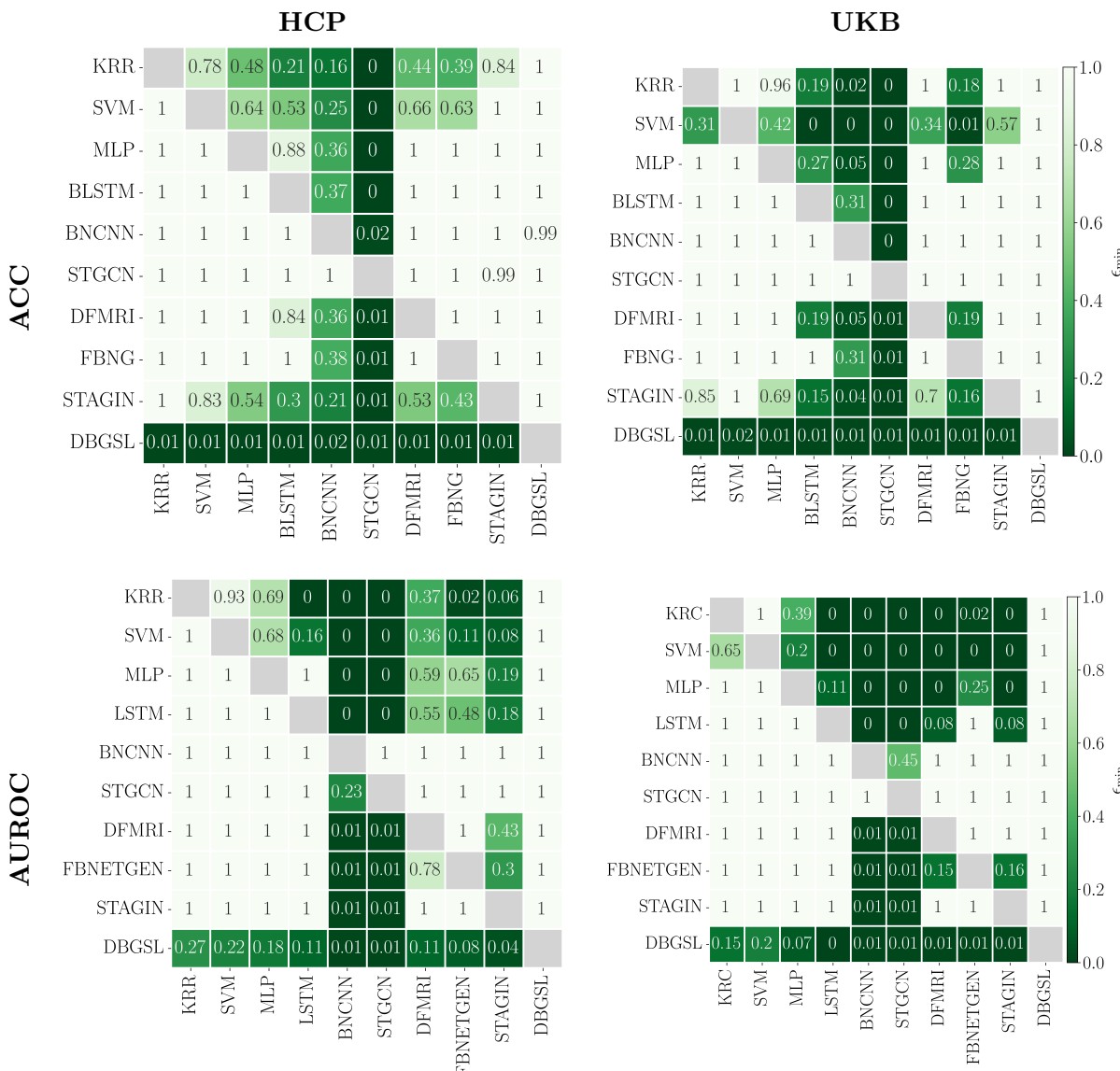

Figure 4: Almost stochastic order (ASO) test scores for the sex classification task on HCP-Test and HCP-Task. ASO scores are expressed as $\epsilon_{\min}$ at $\alpha = 0.05$ significance level adjusted for multiple comparisons using the Bonferroni correction. Read from row to column e.g. for HCP-Rest accuracy (top left) DBGSL (row) is stochastically dominant over STAGIN (column) with $\epsilon_{\min}$ of 0.01.

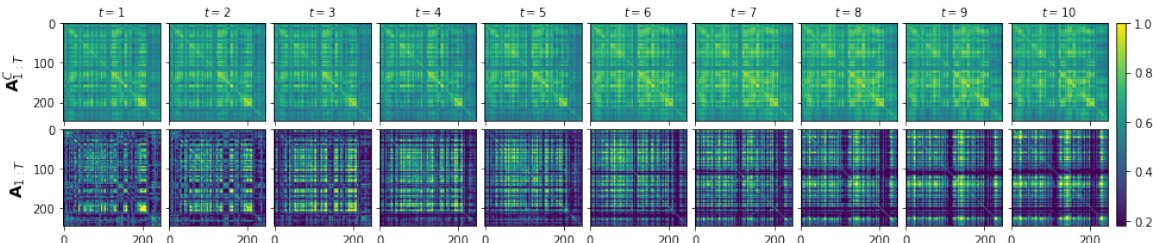

Figure 5: Dynamic FC matrix calculated using Pearson correlation $\mathbf{A}^C_{1:T}$ (normalized to $[0, 1]$) and dynamic adjacency matrix learnt by DBGSL $\mathbf{A}_{1:T}$, using the same BOLD signals from HCP-Rest and a window size and stride of $P = 50$ and $S = 3$, respectively.

## E.1. Sex-discriminative brain regions

We provide further details on the sex-discriminative brain region scores from Figure 2 in Tables 4-9. All brain regions and their respective MNI coordinates are taken from the Brainnetome atlas (Fan et al., 2016). Brain regions are further grouped into intrinsic connectivity networks from Yeo et al. (2011) as well as lobe and gyrus (the outermost layer of the brain).

| Lobe | Gyrus (hemisphere) | Region | MNI (x, y, z) | Score |
|---|---|---|---|---|
| Limbic lobe | Cingulate gyrus (left) | A23d dorsal area 23 | -4, -39, 31 | 0.93 |
| Frontal lobe | Middle frontal gyrus (left) | A8vl ventrolateral area 8 | -33, 23, 45 | 0.92 |
| Frontal lobe | Superior frontal gyrus (right) | A10m medial area 10 | 8, 58, 13 | 0.87 |
| Frontal lobe | Superior frontal gyrus (right) | A9l lateral area 9 | 13, 48, 40 | 0.85 |
| Temporal lobe | Middle temporal gyrus (right) | A21r rostral area 21 | 51, 6, -32 | 0.83 |
| Temporal lobe | Posterior superior temporal sulcus (left) | rpSTS rostroposterior superior temporal sulcus | -54, -40, 4 | 0.82 |
| Temporal lobe | Middle temporal gyrus (left) | A21c caudal area 21 | -65, -30, -12 | 0.81 |
| Temporal lobe | Superior temporal gyrus (right) | A22r rostral area 22 | 56, -12, - 5 | 0.72 |
| Frontal lobe | Inferior frontal gyrus (right) | A45c caudal area 45 | 54, 24, 12 | 0.72 |
| Frontal lobe | Orbital gyrus (left) | A12/47o orbital area 12/47 | -36, 33, -16 | 0.69 |
| Parietal lobe | Precuneus (left) | A31 Area 31 (Lc1) | -6, -55, 34 | 0.23 |
| Limbic lobe | Cingulate gyrus (right) | A32sg subgenual area 32 | 5, 41, 6 | 0.23 |

Table 4: Sex-discriminative brain region scores (normalized to [0, 1]) in the default mode network (DMN) for HCP-Rest (top left Figure 2).

| Lobe | Gyrus (hemisphere) | Region | MNI (x, y, z) | Score |
|---|---|---|---|---|
| Temporal lobe | Superior temporal gyrus (left) | A22c caudal area 22 | -62, -33, 7 | 0.83 |
| Parietal lobe | Inferior parietal lobule (right) | A40rv rostroventral area 40 (PFop) | 55, -26, 26 | 0.79 |
| Frontal lobe | Superior frontal gyrus (right) | A6m medial area 6 | 7, -4, 60 | 0.78 |
| Parietal lobe | Postcentral gyrus (right) | A2 area 2 | 48, -24, 48 | 0.77 |
| Frontal lobe | Precentral gyrus (left) | A4t area 4 (trunk region) | -13, -20, 73 | 0.77 |
| Frontal lobe | Precentral gyrus (left) | A4ul area 4 (upper limb region) | -26, -25, 63 | 0.73 |
| Parietal lobe | Postcentral gyrus (left) | A1/2/3tru area 1/2/3 (trunk region) | -21, -35, 68 | 0.71 |
| Parietal lobe | Postcentral gyrus (right) | A1/2/3tonIa area 1/2/3 (tongue and larynx region) | 56, -10, 15 | 0.70 |
| Temporal lobe | Superior temporal gyrus (right) | TE1.0 and TE1.2 | 51, -4, -1 | 0.22 |

Table 5: Sex-discriminative brain region scores (normalized to [0, 1]) in the somatomotor network (SMN) for HCP-Rest (top middle Figure 2).

| Lobe | Gyrus (hemisphere) | Region | MNI (x, y, z) | Score |
|---|---|---|---|---|
| Temporal lobe | Posterior superior temporal sulcus (left) | Caudoposterior superior temporal sulcus | -52, -50, 11 | 0.72 |
| Limbic lobe | Cingulate gyrus (right) | A24cd caudodorsal area 24 | 4, 6, 38 | 0.70 |
| Frontal lobe | Inferior frontal gyrus (left) | A44v ventral area 44 | -52, 13, 6 | 0.55 |
| Frontal lobe | Inferior frontal gyrus (left) | A44op opercular area 44 | -39, 23, 4 | 0.51 |
| Limbic lobe | Cingulate gyrus (right) | A32p pregenual area 32 | 5, 28, 27 | 0.48 |
| Frontal lobe | Inferior frontal gyrus (right) | A44v ventral area 44 | 54, 14, 11 | 0.47 |
| Insular lobe | Insular gyrus (left) | dIa dorsal agranular insula | -34, 18, 1 | 0.46 |
| Frontal lobe | Precentral gyrus (right) | A4tl area 4 (tongue and larynx region) | 54, 4, 9 | 0.45 |

Table 6: Sex-discriminative brain region scores (normalized to [0, 1]) in the ventral attention network (VAN) for HCP-Rest (top right Figure 2).

| Lobe | Gyrus (hemisphere) | Region | MNI (x, y, z) | Score |
|---|---|---|---|---|
| Temporal lobe | Parahippocampal gyrus (right) | TH area TH (medial PPHC) | 19, -36, -11 | 0.92 |
| Occipital lobe | Medioventral occipital cortex (left) | vmPOS ventromedial parietooccipital sulcus | -13, -68, 12 | 0.91 |
| Occipital lobe | Medioventral occipital cortex (left) | rCunG rostral cuneus gyrus | -5, -81, 10 | 0.88 |
| Occipital lobe | Medioventral occipital cortex (right) | rLinG rostral lingual gyrus | 18, -60, -7 | 0.58 |
| Occipital lobe | Lateral occipital cortex (right) | OPC occipital polar cortex | 22, -97, 4 | 0.56 |
| Parietal lobe | Inferior parietal lobule (left) | A39c caudal area 39 (PGp) | -34, -80, 29 | 0.55 |
| Limbic lobe | Cingulate gyrus (right) | A23v ventral area 23 | 9, -44, 11 | 0.54 |
| Parietal lobe | Precuneus (right) | dmPOS dorsomedial parietooccipital sulcus (PEr) | 16, -64, 25 | 0.50 |
| Occipital lobe | Medioventral occipital cortex (right) | vmPOS ventromedial parietooccipital sulcus | 15, -63, 12 | 0.40 |
| Temporal lobe | Fusiform gyrus (right) | A37mv medioventral area 37 | 31, -62, -14 | 0.38 |
| Temporal lobe | Parahippocampal gyrus (left) | TL area tl (lateral PPHC, posterior parahippocampa) | -28, -32, -18 | 0.37 |
| Temporal lobe | Fusiform gyrus (right) | A37lv lateroventral area 37 | 43, -49, -19 | 0.37 |
| Occipital lobe | Medioventral occipital cortex (right) | rCunG rostral cuneus gyrus | 7, -76, 11 | 0.36 |
| Occipital lobe | Lateral occipital cortex (right) | iOccG inferior occipital gyrus | 32, -85, -12 | 0.36 |
| Temporal lobe | Parahippocampal gyrus (right) | TL area TL (lateral PPHC, posterior parahippocamp) | 30, -30, -18 | 0.36 |

Table 7: Sex-discriminative brain region scores (normalized to [0, 1]) in the visual network (VSN) for HCP-Task (bottom left Figure 2).

| Lobe | Gyrus (hemisphere) | Region | MNI (x, y, z) | Score |
|---|---|---|---|---|
| Subcortical nuclei | Thalamus (right) | mPMtha pre-motor thalamus | 12, -14, 1 | 0.72 |
| Subcortical nuclei | Thalamus (right) | mPFtha medial pre-frontal thalamus | 7, -11, 6 | 0.66 |
| Subcortical nuclei | Thalamus (right) | cTtha caudal temporal thalamus | 10, -14, 14 | 0.65 |
| Insular lobe | Insular gyrus (left) | vIa ventral agranular insula | -32, 14, -13 | 0.61 |
| Subcortical nuclei | Basal ganglia (left) | vCa central caudate | -12, 14, 0 | 0.60 |
| Subcortical nuclei | Basal ganglia (left) | vmPu ventromedial putamen | -23, 7, -4 | 0.46 |
| Subcortical nuclei | Basal ganglia (right) | dlPu dorsolateral putamen | 29, -3, 1 | 0.45 |
| Subcortical nuclei | Thalamus (right) | Otha occipital thalamus | 13, -27, 8 | 0.41 |
| Limbic lobe | cingulate gyrus (left) | A24rv rostroventral area 24 | -3, 8, 25 | 0.36 |

Table 8: Sex-discriminative brain region scores (normalized to [0, 1])in the subcortical network (SCN) for HCP-Task (bottom middle Figure 2).

| Lobe | Gyrus (hemisphere) | Region | MNI (x, y, z) | Score |
|---|---|---|---|---|
| Parietal lobe | Superior parietal lobule (right) | A5l lateral area 5 | 35, -42, 54 | 0.92 |
| Frontal lobe | Precentral gyrus (right) | A6cvl caudal ventrolateral area 6 | 51, 7, 30 | 0.51 |
| Parietal lobe | Superior parietal lobule (left) | A7c caudal area 7 | -15, -71, 52 | 0.50 |
| Frontal lobe | Superior frontal gyrus (left) | A6dl dorsolateral area 6 | -18, -1, 65 | 0.42 |
| Temporal lobe | Inferior temporal gyrus (right) | A37elv extreme lateroventral area 37 | 53, -52, -18 | 0.42 |
| Parietal lobe | Superior parietal lobule (right) | A7r rostral area 7 | 19, -57, 65 | 0.41 |
| Parietal lobe | Inferior parietal lobule (right) | A40rd rostrodorsal area 40 (PFt) | 47, -35, 45 | 0.40 |
| Temporal lobe | Middle temporal gyrus (right) | A37dl dorsolateral area 37 | 60, -53, 3 | 0.40 |

Table 9: Sex-discriminative brain region scores (normalized to [0, 1]) in the dorsal attention network (DAN) for HCP-Task (bottom right Figure 2).

