# OpenReview forum: "DBGSL: Dynamic Brain Graph Structure Learning"
_MIDL.io/2023/Conference — MIDL 2023 Poster_

### Official Review · Reviewer_WRy2 · 2023-02-02

**Confidence:** 4
**Preliminary Rating:** 5
**Recommendation:** Oral

**Summary:**

The author presents a dynamical GNN framework to classify brain regions dynamically from fMRI time series. They apply it to sex classification from brain signals only with a state-of-the-art accuracy. This work is interesting as it shows that dynamics should be taken into account in fMRI data (and possibly other types of measurements) for a precise learning task, suggesting that the dynamical nature of the brain plays a crucial role in determining its properties.

**Strengths:**

The presented method, although technical, is clear and complete despite the paper's length limitations. The method is compared against many recent other ones and shows a significant improvement, making the presented method stand out in the context. The potential applications of this method to other datasets or classification tasks are also clear and exciting.

**Weaknesses:**

There are not many weaknesses in the paper I could find, except the analysis section. Fig. 2 is not referenced much in the text and in my opinion, does not bring anything very relevant to the reader, except that these matrices are different across time, which we already know. I would remove this figure (or move to the appendix) and expand on Fig.3 as follow. The plots are first of all a little small and the brain regions are not highlighted, hence it is difficult for the non-expert to see that the colours on the brains correspond well to the brain regions mentioned in the title. It would definitely improve this section if this figure would be made more clear in the comparison of the results and the real brain regions to better support the claims in the text.

**Deanonymize Review:**

no

**Detailed Comments:**

above eq. (8), there is a missing part in the equation D_t=

**Paper Type:**

both

**Questions To Address In The Rebuttal:**

I find this work quite interesting and I wondered how your work fits within the larger topic of temporal graph learning (see for example a recent workshop https://sites.google.com/view/tglworkshop2022/home) and temporal networks in general. I am not so familiar with these topics but I know there have been a lot of works on these related to various applications. I am wondering if any of these developed methods could be used in your context to further improve the interpretability of your results to better understand the brains' function from these datasets.

Also, there are more and more works on higher-order interactions in the brain, would you think that it could be interesting to replace the graph structure with a higher-order version of graphs to possibly further improve your method?

---

### Official Review · Reviewer_LV9H · 2023-02-04

**Confidence:** 4
**Preliminary Rating:** 5
**Recommendation:** Oral

**Summary:**

The authors presented an end-to-end framework able to learn a dynamic graph structures in multivariate time series and extract features from it using a graph neural network such that the prediction error in a downstream task is minimized. They tested their model for sex classification using fMRI data and significantly outperformed the state-of-the art.

**Strengths:**

To my knowledge coupling a dynamic graph structure learning with a GNN to extract relevant features for the downstream task is not trivial. The authors proved that such an approach can be successfully implemented and used for fMRI data. Moreover, they highlighted the interpretability component of their model. Thus, I consider the paper to bring a solid methodological and application contribution valuable for the community

The authors did an extensive baseline comparison, with several methods and described the training details. The model performance was properly evaluated using multiple comparison correction in the significance analysis. Moreover, to compare the different models they used multiple seeds and the almost stochastic order (ASO) test.

Overall the paper is  well structured and language is clear and it seems like the code is going to be publicly available.


**Weaknesses:**

I would have liked to see how the model performs in another task besides sex classification. The addition of an additional task, perhaps predicting age, or another variable available in the dataset would further strengthen the paper.

Despite the extensive baseline comparison I think there is a case missing in which a dynamic graph structure is learnt. For example, an adjacency matrix could be computed using pearson’s correlation coefficient at each window and then fed to SVM/KRR (since they seem to be the baseline top-performing models in this task).

Overall the paper cites properly the state of the art, and considering the wide range of technical aspects that are covered in this work it is not an easy task. However, in the dynamic graph structure learning there are some papers that could be added to better contextualize the state and previous work on dynamic graph structure learning.

Kalofolias, V., Loukas, A., Thanou, D., & Frossard, P. (2017). Learning time varying graphs. 2017 IEEE International Conference on Acoustics, Speech and Signal Processing (ICASSP), 2826–2830. https://doi.org/10.1109/ICASSP.2017.7952672

Kazemi, S. M., Goel, R., Jain, K., Kobyzev, I., Sethi, A., Forsyth, P., & Poupart, P. (2020). Representation learning for dynamic graphs: A survey. Journal of Machine Learning Research, 21, 1–73.


**Deanonymize Review:**

no

**Detailed Comments:**

Figure quality: 2,3, 5. It would be good to have these images with a higher quality. When printed or zoomed in the resolution is poor.

The notation for the hidden states is a bit confusing. H is used for both the hidden states of the dynamic graph classifier and also for the hidden states of the dynamic graph learner but there is no indication that are two different objects.


**Paper Type:**

methodological development

**Questions To Address In The Rebuttal:**

Given the fact that linear models (SVM/KRR) outperform most of deep learning approaches I consider that it could be a good addition to the benchmark to test how SVM/KRR perform when the input is a series of FC computed at each snapshot using pearson’s correlation coefficient. In this case, the window size and padding would be hyperparameters that you need to select, maybe using a cross-validation strategy. Personally, I consider that if the presented model (DBGSL) still outperforms this baseline it would strengthen the paper even more and also highlight that it is not only the dynamical aspect of the graph but also how this graph is computed that is important. Do you consider that such a test, or a similar one, could be incorporated in the paper?

---

### Official Review · Reviewer_TtE2 · 2023-02-04

**Confidence:** 4
**Preliminary Rating:** 4
**Recommendation:** Poster

**Summary:**

This work presents an end-to-end graph neural network framework for performing classification from dynamic functional connectivity data. The deep brain graph structure learning model (DBGSL) operates on dynamically evolving graph adjacency matrices inferred via spatial attentional modeling of brain region embeddings learned from fMRI time-series data. Additionally, DBGSL also introduces temporal attention mechanisms and learnable edge sparsity to improve the classification performance and aid interpretability.

Evaluation is performed on a sex classification task on the Human Connectome Project data on both resting-state and task fMRI data against several deep learning and machine learning baselines designed for dynamic and static connectivity. Finally, they perform a qualitative analysis of the learnt graphs to highlights brain regions that are most predictive of the classification labels.

**Strengths:**

The proposed end-to-end framework for graph structure learning by leveraging spatial and temporal attention in graph neural networks is an interesting methodological contribution. The presentation of the method is clear and the experimental validation performed is quite thorough. Extensive comparisons have been performed on several state-of-the-art baselines and ablation studies on key hyper-parameters, against which the framework provides improved performance. Relevant implementation details are provided and it seems like the code will be made available, which is a big plus for reproducibility.

**Weaknesses:**

1. Some of the claims of the paper seem a bit overstated. For example, in the contributions section, the authors state that "the first an end-to-end trainable GNN-based model able to learn task-specific dynamic brain graphs from fMRI data in a supervised framework". Some of the baselines that are compared against such as (Gadgil et al 2020) or (Kim et al 2021) are also end-to-end graph neural networks that can perform supervised learning.

2. From the description provided in the interpretability analysis, it is unclear whether the analysis is being performed using score vectors from training/test/all subjects. Has any analysis been performed across the two groups? Are the highlighted regions consistently recovered across different sub-samples of the data for each group?

**Deanonymize Review:**

no

**Detailed Comments:**

A recommended proofread to fix typos and grammar:

"the first an end-to-end trainable " --> the first end to end


**Paper Type:**

methodological development

**Questions To Address In The Rebuttal:**

Please refer to the comments in the weaknesses section. Specifically, it would be useful to know whether the method uncovers consistent differences (in activation patterns) across the population that are particularly useful for the classification task.

---

### Meta-Review · Area_Chair_CPMD · 2023-02-23

**Recommendation:** Accept (Poster)
**Confidence:** 4

**Metareview:**

The paper proposes an approach to jointly learn graph structure and perform a classification task, on dynamic brain functional connectivity data. The experiments are carefully conducted and the method is original. It has applications beyond brain imaging and is therefore probably interesting to a larger audience. The authors have engaged thoughtfully with the reviews.

Pros:
 - Good validation with respect to baselines, including care of random seeds and multiple comparisons correction
 - Application shown to both resting-state and task fMRI data

Cons:
- the pretext task chosen (sex classification) is not particularly impactful for scientific discovery. ConnectomeDB has many other behaviour variables of greater interest (Cognition - NIH toolbox, Sensory, etc.), which could e.g. be discretized, or indeed could be used directly with a regression loss.
- Benchmarking is lacking comparison with linear methods which are often good performers (for classification, not necessarily structure learning) - high dimensionality (quadratic scaling in the number of regions) is not necessarily an issue as there is much redundancy in the signal and the connectivity matrices themselves, so simple mass-univariate feature selection would solve this.